# Species-specific enamel differences in hardness and abrasion resistance between the permanent incisors of cattle (*Bos primigenius taurus*) and the ever-growing incisors of nutria (*Myocastor coypus*)

**Valentin L. Fischer**[1], **Daniela E. Winkler**[2,3], **Robert Głogowski**[4], **Thomas Attin**[5], **Jean-Michel Hatt**[1], **Marcus Clauss**[1] *, **Florian Wegehaupt**[5]

**1** Vetsuisse Faculty, Clinic for Zoo Animals, Exotic Pets and Wildlife, University of Zurich, Zurich, Switzerland, **2** Department of Natural Environmental Studies, Graduate School of Frontier Sciences, University of Tokyo, Kashiwa, Chiba, Japan, **3** Institute of Geosciences, Johannes Gutenberg University Mainz, Mainz, Germany, **4** Institute of Animal Science, Warsaw University of Life Sciences, Warsaw, Poland, **5** Division of Preventive Dentistry and Oral Epidemiology, Centre of Dental Medicine, University of Zurich, Zurich, Switzerland

* mclauss@vetclinics.uzh.ch

## Abstract

Hypselodont (ever-growing) teeth of lagomorphs or rodents have higher wear rates (of a magnitude of mm/week), with compensating growth rates, compared to the non-ever-growing teeth of ungulates (with a magnitude of mm/year). Whether this is due to a fundamental difference in enamel hardness has not been investigated so far. We prepared enamel samples (n = 120 per species) from incisors of cattle (*Bos primigenius taurus*) and nutria (*Myocastor coypus*, hypselodont incisors) taken at slaughterhouses, and submitted them to indentation hardness testing. Subsequently, samples were split into 4 groups per species (n = 24 per species and group) that were assessed for abrasion susceptibility by a standardized brush test with a control (no added abrasives) and three treatment groups (using fine silt at 4 ±1 μm particle size, volcanic ash at 96 ±9 μm, or fine sand at 166 ±15 μm as abrasives), in which enamel abrasion was quantified as height loss by before-and-after profilometry. The difference in enamel hardness between the species was highly significant, with nutria enamel achieving 78% of the hardness of cattle enamel. In the control and the fine sand group, no enamel height loss was evident, which was attributed to the *in vitro* system in the latter group, where the sand particles were brushed out of the test slurry by the brushes' bristles. For fine silt and volcanic ash, nutria enamel significantly lost 3.65 and 3.52 times more height than cattle. These results suggest a relationship between enamel hardness and susceptibility to abrasion. However, neither the pattern within the species nor across the species indicated a monotonous relationship between hardness and height loss; rather, the difference was due to qualitative step related to species. Hence, additional factors not measured in this study must be responsible for the differences in the enamel's susceptibility to abrasion. While the *in vitro* brush system cannot be used to rank abrasive test substances in terms of their abrasiveness, it can differentiate abrasion susceptibility in

**Data Availability Statement:** All relevant data are within the manuscript and its Supporting Information files.

**Funding:** DEW was supported by a European Research Council (ERC) under the European Union's Horizon 2020 research and innovation program (ERC CoG grant agreement no. 681450 to Thomas Tütken) and a Postdoctoral fellowship from the Japan Society for the Promotion of Science (KAKENHI Grant No.20F20325).

**Competing interests:** The authors have declared that no competing interests exist.

dental tissue of different animal species. The results caution against considering enamel wear as a similar process across mammals.

## Introduction

There is a debate whether measures of tooth wear can be considered taxon-free proxies of triggers for wear and hence environmental conditions, in particular in a paleobiological context, or whether the same conditions will lead to different wear patterns in different animal species [1–4] or even in different individuals within a species [5]. Among the many possible factors by which teeth of different species can vary, enamel hardness appears intuitively relevant for the effect of abrasives on dental wear.

There are several potential causes for differences in functional enamel characteristics between species, such as differences in enamel thickness [6,7], incorporation of different minerals [8,9] or different degrees of mineralisation [10], different enamel prism decussation patterns [11] or different enamel *schmelzmuster* [12]. With respect to dental wear, one of the impressive functional differences is the rate of tissue loss in ever-growing (hypselodont or euhypsodont) teeth as compared to non-ever-growing teeth. In hypselodont teeth (both incisors and cheek teeth), wear rates of a magnitude of several millimetres *per week* are known [13–19], with a corresponding, compensating growth rate. By contrast, mammalian herbivores with non-ever-growing teeth yet similar diets show cheek tooth wear rates at a magnitude of millimetres *per year* [13,20–23].

Interpreting such differences as an effect of differences in enamel hardness may be intuitive, both at an inter-specific and intra-specific scale. Support for this hypothesis can be drawn from data on enamel hardness (of unspecified location on the teeth) collected from a variety of sources in Berkovitz and Shellis [24; chapter 3], with lower values for rabbits (*Oryctolagus cuniculus*) than for some large mammals (*Cervus elaphus*, *Ovis aries*, *Bos taurus*, *Equus caballus*). Preliminary data from microhardness testing of rabbit enamel (measured on the chewing surface of both incisors and cheek teeth) point in the same direction [25] when compared, for example, to various enamel hardness measurements in ungulates [26]. However, there is a high variability between methods for enamel hardness measurements and consequent results, and whether samples for measurements were stored under wet or dry conditions prior to measuring has a relevant effect on the results as well [26]; hence, comparability of values from different studies may be compromised to a certain degree. Within a single species, red deer (*Cervus elaphus*), Pérez-Barbería [27] demonstrated no difference in enamel hardness between the sexes. As this finding is paralleled by a generally faster tooth wear in males compared to females [22,28], the author concludes that hardness itself cannot be the main decisive factor for differences in wear in this species, and possibly between female and male ungulates in general.

In the present study, we aimed to test for differences in enamel hardness and wear in two contrasting mammalian species–cattle, with non-ever-growing teeth, and nutria (*Myocastor coypus*), with hypselodont (ever-growing) incisors. We used a test system that is well-established in *in vitro* research on dental tissue in human dentistry, where cattle incisors are used as a model for human teeth [29–33]. In these *in vitro* tests, prepared samples of incisor enamel are submitted to indentation hardness testing, and to abrasion testing in brushing machines under the addition of various abrasive substances. Tissue loss is measured as height difference before and after the brush test using profilometry.

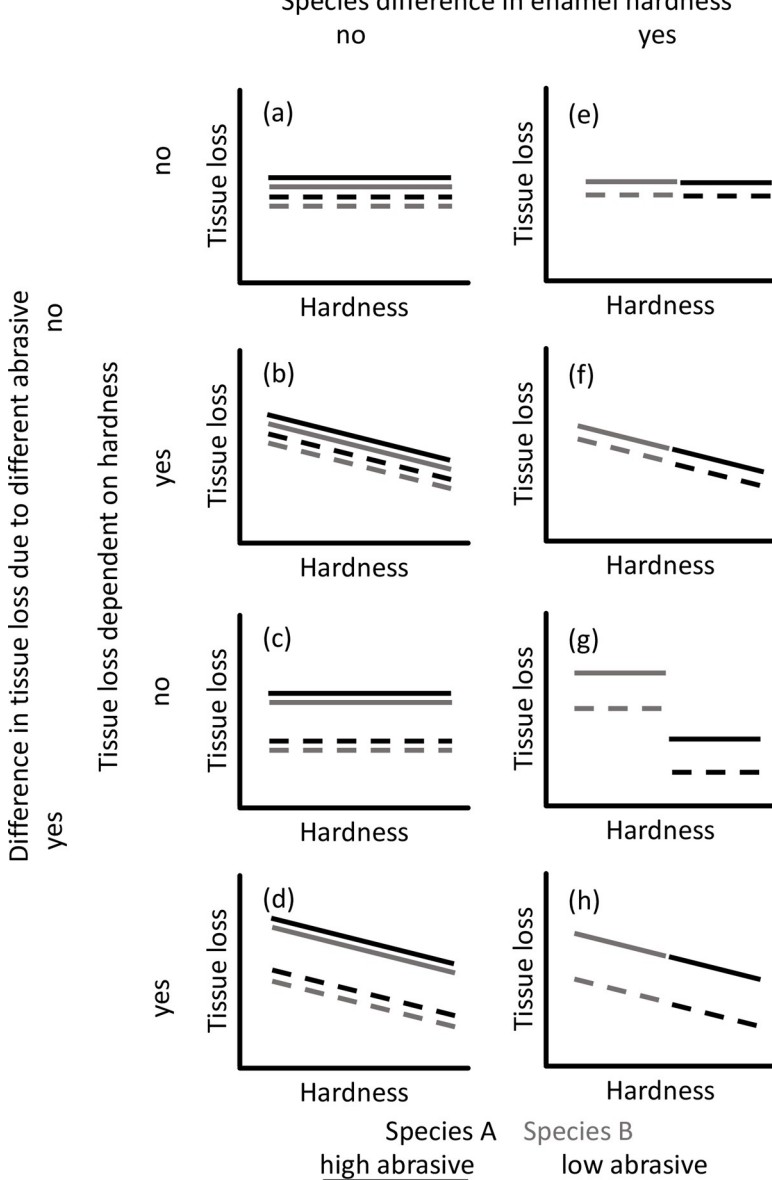

**Fig 1. Schematic representation of potential results of this study depending on different combinations of whether species differ in enamel hardness (columns), and whether different abrasives lead to different tissue loss irrespective of, or in proportion to, hardness.**

Among the theoretically feasible results, we present a combination of eight possible patterns (Fig 1). These patterns derive from the combinations of either no species difference in enamel hardness (Fig 1A–1D) or a distinct difference between the species (Fig 1E–1H); from no difference in tissue loss due to different abrasive substances (Fig 1A, 1B, 1E and 1F) or a clear difference in tissue loss depending on the abrasive used (Fig 1C, 1D, 1G and 1H); and from whether there is no dependence of tissue loss on enamel hardness (Fig 1A, 1C, 1E and 1G) or a decrease in tissue loss at increasing hardness (Fig 1B, 1D, 1F and 1H). Based on the considerations outlined above, we expected a softer enamel in nutria as compared to cattle (Fig 1E-1H). Given previous results on different effects of different abrasives on dental wear patterns [34–37], we

expected tissue loss to differ between the abrasives tested (Fig 1G and 1H), and we finally expected that softer enamel samples would lose more tissue in the standardized test, as demonstrated previously with cattle enamel [38]. Thus, we predicted a pattern as in Fig 1H.

## Materials and methods

We compared the hardness and resistance to *in vitro* height loss of cattle and nutria enamel. Nutria teeth originated from animals raised at a commercial nutria farm in Poland, where they had been fed low energy density diets such as fresh green forage and plant production by-products offered in relatively large amounts. These animals were raised for meat production, and slaughtered according to standard procedures at a Polish slaughterhouse at an age of 10 to 12 months. Cattle teeth originated from animals raised as beef cattle in Switzerland at various farms under unknown conditions, slaughtered according to standard procedures at a Swiss slaughterhouse at an age of 2–3 years. In both species, teeth were extracted from the skulls (that are treated as a waste product in the slaughter process), and stored in water under refrigeration until processing. Sampling of slaughtered animals from regular production systems is not considered an animal experiment and hence not subject to ethical clearance. Note that cattle only have mandibular incisors; in the nutria, both mandibular and maxillary incisors were sampled, but the identity of the animal was not recorded. In other words, it was not possible to link a specific maxillary incisor sample to the mandibular incisor sample of the same individual.

Dental samples from the labial side of the incisors were prepared for hardness measurements and abrasion tests following Attin and Wegehaupt [39]. We took cylinders (inner diameter 3 mm) from teeth with a trephine drill (Komet, Lemgo, Germany). The cylinders were placed in cylindrical aluminium sample moulds (inner diameter 5 mm) and embedded in methylmethacrylate (Paladur, Kulzer, Hanau, Germany). Subsequently, the enamel surface was polished with water-cooled carborundum discs of increasing fineness (800, 1000 and 1200, 2400 and 4000 grit; Water Proof Silicon Carbide Paper, Struers, Erkrath, Germany) with a digitally controlled automatic grinding device (Exact Mikroschleifsystem Mikro 40, Exakt Apparatebau, Norderstedt, Germany) to yield a flat surface.

A total of 120 cattle samples (representing mandibular incisors) and 120 nutria samples (representing equal amounts of maxillary and mandibular incisors) were prepared in this way. All samples underwent hardness testing; for testing the susceptibility to tissue loss under standardized brushing with abrasives, samples of a species were divided into 4 groups of 24 samples each. In doing so, we ensured that in each group, the same proportion of nutria mandibular and maxillary samples were present, and for each species, the whole range of available hardness was covered, to better assess the relationship between hardness and tissue loss as indicated in Fig 1.

We assessed hardness as Knoop microindentation hardness using a 1600–6106 hardness tester (Buehler, Lake Bluff, IL USA). The principle of the measurement is the application of a known force to an indenter for a defined period of time to a section of the sample that is selected by microscopic visual assessment. The recorded size of the inflicted indentation lesion is transformed used a standard equation to a measure of hardness (kgf/mm$^2$, or the Knoop Hardness number HK), with 3 replicate measures per sample.

We assessed abrasion susceptibility using an automated brushing machine [30] applying linear, reciprocating strokes (60 per minute with a load of 300 g). Brushing was performed for a total of 6 hours. A standard toothbrush with medium bristle stiffness was used, with a multi-tufted, flat design, and a new brush was used for every 6-hour-run. For the control group, only artificial saliva prepared according to McDougall [40] was used. For the treatment groups, a

suspension of 3 ml hec-glycerine (mixed at a ratio of 5 parts glycerine and 1 part abrasive) was added per sample and renewed after 3 hours. Three different abrasive treatments were assessed: no abrasive (control), fine silt and fine sand (SCR-Sibelco N.V., Antwerp, Belgium; SIRCON® M500, mean particle size 4 ±1 μm; METTET AF100, mean particle size of 166 ±15 μm), and volcanic ash (Hess pumic mine, Idaho; mean particle size of 96 ±9 μm). For a detailed characterization of these abrasives, see Winkler et al. [37]. In the brush machine, one tray that contains a certain slurry-abrasive combination can harbour two enamel samples. We always filled these two slots with one nutria and one cattle sample, alternating the front/back positions. Also, between runs, we changed to position of the abrasives in the machine, so that each arm of the brush machine acted on a similar number of runs for each abrasive. We applied profilometry to the samples before and after abrasion treatment using a Perthometer S2 (Mahr, Göttingen, Germany) [41]. Markers on the samples are used for an exact positioning of the samples ensuring an exact superimposition of the profiles obtained during the two measurements. The spacing and length of the profiles are 250 and 1500 μm. Average height of enamel loss in comparison to the baseline surface profile is calculated by the manufacturer's software. The detection limit of the here used setup has been reported as 0.105 μm [42]. One sample of nutria enamel of the fine silt group and one sample of cattle enamel of the volcanic ash group had to be excluded from the analyses, because the enamel had been worn off completely, exposing the underlying dentin and thus making measurements of enamel height loss unreliable. In all other cases, the enamel layer submitted to testing was not worn off completely.

We first compared the hardness of nutria maxillary and mandibular incisors using the Student t-test (after confirming normal distribution by Shapiro-Wilk test and equal variances by Levene's test). As there was no difference (see below), nutria samples were subsequently treated as one unit. Differences in enamel hardness between cattle and nutria were assessed using a Weltch t-test (after confirming normal distribution by Shapiro-Wilk test but detecting unequal variances by Levene's test). To assess the effect of hardness on enamel height loss across the species, a general linear model (GLM) was performed, using (log-transformed or ranked) height loss of a sample as the dependent variable, (log-transformed or ranked) sample hardness as the independent variable, and species and abrasive as cofactors. All 2-way interactions were included in the model but dropped if not significant. Model assumptions were confirmed by submitting model residuals to a Shapiro-Wilk normality test. Analyses were performed in R [43], with the significance level set to 0.05.

## Results

There was no significant difference in enamel hardness between maxillary and mandibular nutria incisors ($t = 1.1$, $P = 0.275$) (Table 1). By contrast, cattle incisors had significantly harder

**Table 1. Mean ± standard deviation (range) Knoop indentation hardness of incisor enamel of nutria (*Myocastor coypus*) and cattle (*Bos primigenius taurus*).**

| Species | Jaw | n | Enamel hardness (kgf/mm$^2$) |
|---|---|---|---|
| Nutria | maxilla | 60 | 230 ±30[a] (155–294) |
| | mandible | 60 | 224 ±30[a] (150–284) |
| | combined | 120 | 224 ±30[A] (150–294) |
| Cattle | mandible | 120 | 291 ±22[B] (216–341) |

[a]means not significantly different by Student t-test ($P = 0.275$).
[AB]means significantly different by Weltch t-test ($P < 0.001$).

**Table 2. Mean ± standard deviation (range) height loss incisor enamel of nutria (*Myocastor coypus*) and cattle (*Bos primigenius taurus*) in a standardized brush test with different treatments (n = 24 per treatment and species, except nutria fine silt n = 23).**

| Species | Height loss (μm) | | | |
| --- | --- | --- | --- | --- |
| | control | fine sand | volcanic ash | fine silt |
| Nutria | 0.01 ± 0.01 (0.00–0.04) | 0.03 ± 0.05 (0.00–0.19) | 10.27 ± 3.88 (4.44–22.53) | 26.87 ± 13.47 (11.05–66.88) |
| Cattle | 0.02 ± 0.02 (0.00–0.05) | 0.02 ± 0.02 (0.00–0.07) | 2.92 ± 1.14 (0.98–5.00) | 7.35 ± 3.09 (2.21–13.83) |

enamel than nutria incisors ($t = 18.8$, $P < 0.001$) (Table 1). The average difference was 64 kgf/mm$^2$; the mean hardness of nutria enamel was 78% of cattle enamel.

For the control group as well as the fine sand treatment, height loss was below the reliable detection threshold in both species (Table 2). Therefore, these groups were analyzed separately from the fine silt and volcanic ash treatment.

Comparing the fine silt and volcanic ash groups by GLM had to be performed on log-transformed data. None of the two-way interactions were significant ($P > 0.131$). In the model without the interactions, there was no effect of hardness, but significant effects of species (with nutria showing more height loss) and abrasive (with more height loss on fine silt) (Table 3). The resulting data pattern (Fig 2A) did not resemble the prediction of Fig 1H but the outcome depicted as Fig 1G. The mean enamel height loss ratio of nutria/cattle was very similar between the two abrasives, at 3.65 for fine silt and 3.52 for volcanic ash.

Comparing the control and fine sand groups by GLM had to be performed on ranked data (due to several cases of zero height loss). It must be remembered that the height loss measurements were below the reliability threshold, making this model particularly exploratory. The difference between the species was significant, with cattle having higher ranks, i.e. more height loss, even though the nutria samples had the larger range in enamel height loss (Table 2). Hardness had a significant effect in this model, with less height loss in harder samples (Table 3); however, this was only evident in cattle (Fig 2B), and hence the species × hardness interaction was significant (Table 3). There was no difference between the control and the fine sand (Table 3).

## Discussion

The present study clearly indicates differences in enamel hardness and enamel susceptibility to abrasion between two species, selected as representatives for non-ever-growing and ever-growing teeth. While the species difference in susceptibility to abrasion matches our expectations based on the species difference in enamel hardness, a monotonous relationship between the two measures was absent across and within species, cautioning against a simplistic assumption of causation. Several peculiarities of the results must be ascribed to the *in vitro* measurements and should not be transferred directly to the *in vivo* situation.

**Table 3. Results of general linear models assessing the effects of species, abrasive and enamel hardness on enamel height loss after a standardized brush test.**

| Groups | Intercept | | Species | | Abrasive | | Hardness | | 2-way interactions |
| --- | --- | --- | --- | --- | --- | --- | --- | --- | --- |
| | *t* | *P* | *t* | *P* | *t* | *P* | *t* | *P* | |
| Volcanic ash & fine silt* | 0.680 | 0.498 | 9.78 | <0.001 | 10.63 | <0.001 | 0.18 | 0.860 | n.s. |
| Control & fine sand˚ | 5.42 | <0.001 | 3.86 | <0.001 | 1.70 | 0.093 | -3.76 | <0.001 | Species × Hardness $t = 2.32$, $P = 0.023$ |

*log-transformed data.
˚ranked data.

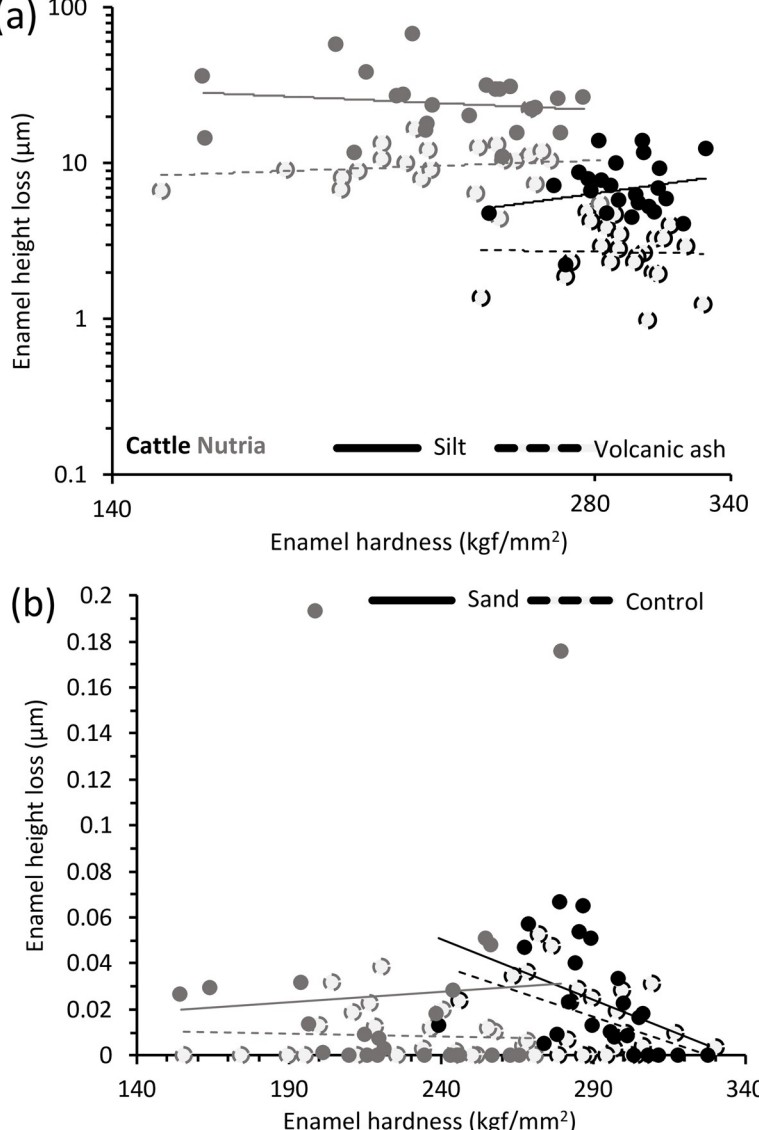

**Fig 2.** Relationship of enamel hardness and enamel height loss in cattle (*Bos primigenius taurus*, black data) and nutria (*Myocastor coypus*, grey data) in (a) samples exposed to fine silt or volcanic ash; (b) samples of the control group and samples exposed to fine sand. For statistics, see Table 3. Note the log-log-scale in (a) and the trendlines that are generally not significant but only added to emphasize the data pattern.

The hardness results of the present study can be considered reliable, especially because all samples were kept under constant hydration after the death of the animal. Differences in exsiccation status are relevant when measuring enamel hardness [26] but can be excluded due to the sample storage in the present study. The magnitude of the difference in our data, where nutria enamel had 78% of the hardness of cattle enamel, mirrors the difference in the data collection of Berkovitz and Shellis [24; chapter 3], where rabbit enamel has 80% of the hardness of cattle enamel. Nevertheless, a larger catalogue of enamel hardness, and possibly other characteristics, for mammal species would be desirable.

In the present study, enamel samples were taken from the labial side of the incisors, which is, in the rodent, the only side covered in enamel [24,44,45]. For this material, we consider our

indentation hardness measures as representative for the whole sample, so that a lack of hardness-height loss relationship cannot be ascribed to non-representative hardness measurements. When measuring the indentation hardness of a composite sample of enamel and dentin of an occlusal surface, it is known that measures taken on the enamel close to the enamel-dentin junction typically show lower hardness than samples measured further away from that junction [46]. The hardness measurement and the wear challenge were applied to enamel from the (labial) side of the tooth. In live animals, wear will occur at the occlusal surface of the tooth–in nutria, mainly due to contact with the other incisor, and in cattle (with no maxillary incisors) due to contact with food. Therefore, the wear measured our in vitro system is unlikely to resemble wear in live animals. Notably, the enamel microstructure, which is particularly complex in rodents [47], will serve to reinforce the enamel in the functional chewing direction and not necessarily against unnatural abrasion from the side.

Based on *in vivo* experiments comparing dental microwear texture from animals ingesting the same abrasives as those used in the *in vitro* brush test of the present study [34,37], as well as on *in vivo* studies with animals receiving diets with or without included sand [14,15,17,20], we had expected tissue loss to be highest in the fine sand, intermediate in the volcanic ash, and lowest in the fine silt samples. However, the results showed the opposite ranking, with the fine sand treatment being indistinguishable from the control treatment. We do not consider this an indication for a lack of abrasiveness of the sand, but as a peculiarity of our *in vitro* brush test system. In the *in vitro* test, the size relationship of the abrasive test substance and the distance between the bristles of the test brushes will influence the efficacy of the test substance. Particles of a distinctively smaller size than the bristle distance will be moved along with the slurry across the enamel samples, whereas larger particles may be selectively removed from the slurry by the brushes. Indeed, the visual impression was that the brushes quickly removed the fine sand (the largest abrasive particles in our experiment) from the whole of the slurry and pushed them to the front and back edges of the test trays that contained the enamel samples. This was not observed for the finer volcanic ash or fine silt. Hence, a brush test should not be considered a suitable tool to rank the effect of abrasives, because of the possible interaction of the abrasives with the test system. Therefore, we also do not claim that a concentration of fine silt will lead to more enamel loss *in vivo* than the same concentration of volcanic ash. Nevertheless, the method can yield general results on the susceptibility of material to abrasion. In particular, the fact that the ratio of enamel height loss in our test species nutria:cattle was, at 3.52 for volcanic ash and 3.65 for fine silt, comparable between two smaller-sized abrasives, suggests that a susceptibility ranking of enamel to abrasion can be reliably produced when a test abrasive suitable for the method is chosen.

Additionally, when the fine silt and the fine sand were compared directly in feeding experiments *in vivo*, no difference in macroscopic tooth wear between these abrasives was detected [19,21]. In terms of microwear texture, the fine silt leads to a pattern of 'enamel polishing' with smoother surfaces compared to controls, whereas the fine sand leads to a pattern of 'enamel scratching' with rougher surfaces compared to controls [34,37]. Together with the findings of the present study, these observations suggest that different microscopic surface patterns may be related to similar tissue loss, with no clear link between the microscopic and the macroscopic pattern. To further elucidate the relationship between microscopic and macroscopic wear, experiments like the present one might be helpful that also quantify dental microwear texture in enamel samples from different species after exposure to standardized i*n vitro* brush tests.

Although the susceptibility to abrasion was higher in nutria, i.e. the species with the lower enamel hardness, there was no effect of hardness beyond the general species difference in the two treatment groups that yielded reliable enamel height loss measurements. This was in

contrast to our expectation and to previous reports on cattle samples used in a very similar *in vitro* setting [38]. Additionally, a retrospective study in red deer found the expected negative relationship between individual animal's molar enamel hardness and molar wear [46]. By contrast, Muylle et al. [48] found that differences in incisor hardness between horse breeds did not parallel differences in incisor wear, and Pérez-Barbería [27] found that the differences observed in dental wear between female and male red deer were not paralleled by differences in enamel hardness. In these studies, dental wear was also subject to potential other, not studied factors, such as differences in relative food intake or in diet composition. Such uncontrolled factors were excluded in the present *in vitro* study, and yet the expected relationship between hardness and susceptibility to abrasion was not evident beyond the species comparison. Evidently, other characteristics than hardness must be involved in determining enamel's susceptibility to wear. To make progress in this open question, more parallel measurements of hardness in studies focussing on other measures of enamel or dental properties would be required, as also suggested by Pérez-Barbería [27].

The two treatments that hardly caused measurable enamel height loss–the control and the fine sand treatment–showed indication of a hardness-dependent effect in the cattle but not in the nutria samples. Given the caution with which these results need to be interpreted, this finding rather adds to the open questions resulting from our study. An absence of any hardness-related effect in the nutria indicates that it should be some other characteristic of cattle enamel that is responsible for these putative effects.

For the present experiment, we deliberately chose two species for which dental material was relatively easily available (from slaughterhouses), and in which we expected a very distinct difference in hardness due to the non-ever-growing/hypselodont dichotomy outlined in the introduction. For more detailed investigations on the interplay of enamel hardness and abrasion, several considerations–apart from simply increasing the number of investigated species–apply. For example, it might be more interesting to investigate phylogenetically more closely related species, excluding large differences in terms of oral anatomy and chewing physiology, yet still putatively differing in enamel hardness, for example bovids and giraffids that appear particularly vulnerable to dental wear [49]. Additionally, expanding the method to enamel from cheek teeth may be more relevant for non-gnawing mammalian taxa.

In conclusion, we provide clear evidence for a species difference in enamel hardness and enamel susceptibility to abrasion between cattle and nutria. Some of the results lead to recommendations for future work using standardized brush tests, especially with respect to the abrasive substance used–of the substances used here, fine silt is expected to yield the clearest differentiating signal. By contrast, these brush tests are not suited to evaluate the abrasive potential of different-sized abrasives. Factors that determine the susceptibility to wear of an individual enamel sample need to be further elucidated.

## Supporting information

**S1 Table. Original data.** Hardness and tissue loss.
(XLSX)

## Acknowledgments

We thank Andrea Gubler of the Division of Preventive Dentistry and Oral Epidemiology, Centre of Dental Medicine, University of Zurich, for the support of the lab work, and Helder Gomes Rodrigues and Thomas Martin for comments on the initial manuscript.

## Author Contributions

**Conceptualization:** Daniela E. Winkler, Thomas Attin, Jean-Michel Hatt, Marcus Clauss, Florian Wegehaupt.

**Data curation:** Valentin L. Fischer, Daniela E. Winkler, Marcus Clauss, Florian Wegehaupt.

**Formal analysis:** Marcus Clauss.

**Investigation:** Valentin L. Fischer, Daniela E. Winkler.

**Methodology:** Florian Wegehaupt.

**Project administration:** Valentin L. Fischer, Marcus Clauss, Florian Wegehaupt.

**Resources:** Robert Głogowski, Thomas Attin, Jean-Michel Hatt, Florian Wegehaupt.

**Supervision:** Marcus Clauss, Florian Wegehaupt.

**Visualization:** Marcus Clauss.

**Writing – original draft:** Valentin L. Fischer, Marcus Clauss.

**Writing – review & editing:** Daniela E. Winkler, Robert Głogowski, Thomas Attin, Jean-Michel Hatt, Florian Wegehaupt.

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
