## [Decision Letter · Decision Letter 0]

21 Dec 2021

PONE-D-21-35327Species-specific enamel differences in hardness and abrasion resistance between the permanent incisors of cattle (Bos primigenius taurus) and the ever-growing incisors of nutria (Myocastor coypus)PLOS ONE

Dear Dr. Clauss,

Thank you for submitting your manuscript to PLOS ONE. After careful consideration, we feel that it has merit but does not fully meet PLOS ONE’s publication criteria as it currently stands. Therefore, we invite you to submit a revised version of the manuscript that addresses the points raised during the review process.

Both reviewers point out that many specificities of enamel thickness and occlusion differences between the studied species that should be presented and discussed in more details.

We look forward to receiving your revised manuscript.

Kind regards,

Cyril Charles

Academic Editor

PLOS ONE

Journal Requirements:

We thnk Andrea Gubler of the Division of Preventive Dentistry and Oral Epidemiology, Centre of Dental Medicine, University of Zurich, for the support of the lab work. DEW was supported by a European Research Council (ERC) under the European Union’s Horizon 2020 research and innovation program (ERC CoG grant agreement no. 681450 to Thomas Tütken) and a Postdoctoral fellowship from the Japan Society for the Promotion of Science (KAKENHI Grant No. 20F20325).

Reviewers' comments:

Reviewer's Responses to Questions

**Comments to the Author**

1. Is the manuscript technically sound, and do the data support the conclusions?

Reviewer #1: Yes

Reviewer #2: Yes

2. Has the statistical analysis been performed appropriately and rigorously? 

Reviewer #1: Yes

Reviewer #2: Yes

3. Have the authors made all data underlying the findings in their manuscript fully available?

Reviewer #1: Yes

Reviewer #2: Yes

4. Is the manuscript presented in an intelligible fashion and written in standard English?

Reviewer #1: Yes

Reviewer #2: Yes

5. Review Comments to the Author

Reviewer #1: This very interesting paper deals with the influence of enamel hardness on dental wear comparing two mammal species, cattle and nutria, presenting different dental characteristics. This study focusing on incisors provides counter-intuitive but intriguing results showing that dental wear tends to be more species-dependent than hardness-dependent and that other factors must be taken into account, which is discussed accordingly. As a result, this work raises a lot of questions on the differences observed between the two species, and some of them could probably be answered by studying closer taxa having more similar incisor properties (except hardness, and then crown height). My main (but sole) concern is thus the choice of species and of the tooth locus, which should be more accurately explained in relation to the main question of the study based on brachyodont vs hypselodont taxa. I understand that the availability of large samples of cattle and nutria from slaughterhouses could be one of the main criteria for having chosen these two species. However, they present too many different incisor characteristics (some of them are mentioned in the paper) in addition to crown height (e.g. only lower incisors in cattle, only one pair of incisors with only labial enamel in nutria, different enamel thicknesses and microstructures, high iron oxide enrichment in the enamel of nutria, different bite forces and jaw motions) which likely have an impact on wear and render these results more difficult to compare from a biomechanical viewpoint. For these reasons, study of molars (presenting less differences) or comparisons of cattle and nutria with other bovids (or cervids), rodents or with equids (having hypsodont incisors) could have been more appropriate to reduce the potential effect of other factors on wear compared to hardness or crown height. I have also a few minor comments on the manuscript, which deserves to be published pending these modifications.

- L. 71-81: you should precise the tooth loci investigated in the previous publications

- L. 177, 180, 186: write “Shapiro-Wilk”

- L. 313: write “We thank…”

Helder GOMES RODRIGUES

Reviewer #2: This is a well-executed, concise experimental study on enamel abrasion in the evergrowing incisors of Myocastor and the non-evergrowing lower incisors of Bos taurus. The authors have performed enamel hardness measurement by indentation and a brushing experiment with three different abrasive agents plus a control group.

The methodology is well described, the results are clearly outlined and the discussion is supported by the results.

When discussing the different wear rates in the incisors of Myocastor and Bos, the authors might consider the difference in enamel thickness. Rodent incisors generally have a comparatively thin labial enamel cover which together with the underlying softer dentine provides a sharp cutting edge. Enamel microstructure of rodent incisor enamel is among the most complex, if not the most complex, schmelzmuster within mammals, and it is assumed that this is closely related to the high stresses that occur in the enamel during the gnawing process (in order to prevent the enamel from failure). So the thinness of the rodent enamel may play an important role in the higher abrasive rate. Another factor is found in the antagonistic incisor – during the gnawing process, the incisor occlusal surfaces get in contact and therefore experience increased abrasion (self-sharpening mechanism). In the cattle, there are no upper incisors and therefore no tooth-tooth contact. Perhaps the authors can add a couple of sentences to the discussion considering these aspects.

If I got it right, the enamel samples from the nutria incisors were taken from the labial side, and were brushed from that side in the experiment. However, in the living animal enamel abrasion occurs on the occlusal surface of the incisors, which is at an angle to the labial side of the enamel cover. As the authors state in the introduction, enamel hardness is also thought to be dependent of the orientation of the enamel crystallites (e.g. radial enamel of the outer portion in rodent incisors is assumed to be particularly resistant against wear due to the steep inclination of the enamel prisms [wear occurs at a right angle to the c-axes]). When discussing the differences of the experimental enamel abrasion rate, this aspect would be worthwhile to be mentioned.

Line 93: …(Myocastor coypus), with hypselodont (evergrowing) teeth. Replace “teeth” with “incisors”, bescause the cheek teeth of Myocastor are not evergrowing.

6. PLOS authors have the option to publish the peer review history of their article (what does this mean?). If published, this will include your full peer review and any attached files.

Reviewer #1: No

Reviewer #2: **Yes: **Thomas Martin

---

## [Author Response · Author response to Decision Letter 0]

8 Feb 2022

Please see the attached reply letter

---

## [Editor Report · Decision Letter 1]

28 Feb 2022

Species-specific enamel differences in hardness and abrasion resistance between the permanent incisors of cattle (Bos primigenius taurus) and the ever-growing incisors of nutria (Myocastor coypus)

PONE-D-21-35327R1

Dear Dr. Clauss,

We’re pleased to inform you that your manuscript has been judged scientifically suitable for publication and will be formally accepted for publication once it meets all outstanding technical requirements.

Kind regards,

Cyril Charles

Academic Editor

PLOS ONE
---

## [Editor Report · Acceptance letter]

2 Mar 2022

PONE-D-21-35327R1 

Species-specific enamel differences in hardness and abrasion resistance between the permanent incisors of cattle (*Bos primigenius taurus*) and the ever-growing incisors of nutria (*Myocastor coypus*) 

Dear Dr. Clauss:

I'm pleased to inform you that your manuscript has been deemed suitable for publication in PLOS ONE. Congratulations! Your manuscript is now with our production department. 

Kind regards, 

on behalf of

Dr. Cyril Charles 

Academic Editor

PLOS ONE